# Characterization of Testicular Tumor Lesions in Dogs by Different Ultrasound Techniques

**DOI:** 10.3390/ani12020210

**Published:** 2022-01-17

**Authors:** Riccardo Orlandi, Emanuela Vallesi, Cristiano Boiti, Angela Polisca, Paolo Bargellini, Alessandro Troisi

**Affiliations:** 1Anicura Tyrus Clinica Veterinaria, Via Bartocci 1G, 05100 Terni, Italy; riccardo.orlandi@anicura.it (R.O.); emanuela.vallesi@anicura.it (E.V.); paolo.bargellini@anicura.it (P.B.); 2Anicura CMV Clinica Veterinaria, Via G.B. Aguggiari 162, 21100 Varese, Italy; 3Tyrus Science Foundation, Via Bartocci 1G, 05100 Terni, Italy; boiti.cristiano@gmail.com; 4Department of Veterinary Medicine, University of Perugia, Via San Costanzo 4, 06126 Perugia, Italy; 5School of Biosciences and Veterinary Medicine, University of Camerino, Via Circonvallazione 93/95, 62024 Macerata, Italy; alessandro.troisi@unicam.it

**Keywords:** testes, ultrasonography, CEUS, color Doppler, B-flow, canine tumors

## Abstract

**Simple Summary:**

In the present work, we investigated the hypothesis that testicular tumor lesions of dogs could present a specific vascular pattern, which could be detected and differentiated by distinctive ultrasonography techniques. This is a relevant issue since it could contribute to determine their etiology based on the different vascular patterns before dogs undergo surgery. To this end, we implemented a multiple ultrasonographic approach consisting of B-Mode ultrasonography, color Doppler ultrasound, B-flow, and contrast enhanced ultrasound in 27 dogs with different testicular tumors, including leydigomas (*n* = 14), seminomas (*n* = 8), sertoliomas (*n* = 6), and mixed cells (*n* = 5). B-Mode ultrasonography did not differentiate tumor types. Pulsatility and resistive indexes of pampiniform and testicular arteries as assessed by pulse Doppler as well as the presence perilesional and/or perilesional/intralesional blood flow patterns as assessed by color and pulsed Doppler and B-flow differed between tumor types. In conclusion, despite the limited number of cases, we found that testicular tumors of dogs have subtly different vascular patterns. These architectural details are enhanced by multiparametric ultrasonography, which is highly recommended for the identification of their etiology.

**Abstract:**

In this retrospective study, we assessed the accuracy of different blood flow imaging in diagnosing testicular tumor types in dogs. We recruited 27 dogs with leydigomas (14), seminomas (eight), sertoliomas (six), and mixed cells (five) confirmed histopathologically. In intact dogs, Pampiniform plexus and marginal arteries were scanned through pulsed Doppler. Blood flow and presence of intralesional/perilesional arteries were assessed by color and power Doppler, B-flow, and contrast-enhanced ultrasound. Tumor types did not differ by B-Mode ultrasonography characters. Pampiniform and testicular arteries of sertoliomas had higher (*p* < 0.05) pulsatility and resistive indexes. The proportion of leydigomas with a perilesional and/or perilesional/intralesional blood flow pattern detected by color and pulsed Doppler and B-flow was higher (*p* < 0.05) than that of the other tumors counted together. This resulted in a sensitivity of 81.8%, 83.3%, and 85.7%, a specificity of 76.5%, 56.3%, and 73.7%, and a correct classification rate of 78.6%, 67.9%, and 78.8%, respectively. While contrast enhanced ultrasound was highly effective in detecting all tumors, qualitative and quantitative parameters did not contribute to their differential diagnosis. In conclusion, results indicate that different testicular tumor types of dogs have subtly different vascular patterns, a condition that could help in identifying leydigomas.

## 1. Introduction

Testicular tumors represent the most common tumors of the canine male genitalia with a prevalence that ranges from 2% to 60% and an incidence that increases with age [1,2,3]. Among the testicular tumors, seminomas, Sertoli cell tumors, and Leydig cell tumors are the most common [4].

For many years, two-dimensional ultrasonography has been part of routine investigation of the male reproductive tract in both human and veterinary medicine [5,6]. It represents the imaging technique of choice for the detection and characterization of testicular lesions. Even though this diagnostic approach can help in differentiating neoplastic processes from other pathologies, such as orchitis, testicular torsion, and epididymitis, the ultrasonographic changes are not specific enough to identify the different types of testicular tumors [7,8]. In fact, the ultrasonographic features of testicular tumors are extremely variable [9,10]. Therefore, once detected, testicular tumor-like lesions require further characterization.

Recently, color Doppler ultrasonography and contrast enhanced ultrasound (CEUS) have been applied in in human males and dogs to improve the diagnostic accuracy of testicular tumors and help to decide whether sparing surgery or total orchiectomy was indicated [7,11,12,13,14]. In fact, two-dimensional ultrasonography offers valuable, but only structural information about testicular tumors, while color Doppler and CEUS provide information about their real time blood flow [15]. This information is clinically relevant because vascularization and micro-vessel density, i.e., the number of vessels per mm^2^, reflect tumor angiogenesis and are related to its malignancy [16,17]. Many studies have shown that angiogenesis promotes tumor growth by supplying essential oxygen and nutrients to neoplastic cells [17,18,19,20]. At the end of the last decade, the development of the B-flow technique with high-frequency transducers allowed a non-Doppler visualization of blood flow imaging [21] that was considered superior for evaluating complex hemodynamics of peripheral vascular pathologies in human males [22].

It is therefore not surprising that a multiparametric approach consisting of conventional ultrasonography, color Doppler ultrasound, and CEUS is increasingly being used in humans for differentiating benign from malignant intra-testicular lesions [15]. On the contrary, as per our knowledge, such a multiparametric approach has never been tested in the small animal practice. Indeed, the use of reliable diagnostic tools would be relevant for the detection and characterization of testicular tumors that can compromise fertility, especially in stud dogs. We focused our attention on breeding dogs to evaluate the possibility of preserving their fertility. In addition, an early diagnosis of a testicular tumor can also reduce the risk of metastasis, even if that occurs less frequently in dogs than humans [23].

Therefore, in the present study, we evaluated the diagnostic accuracy of two-dimensional ultrasonography combined with Doppler ultrasound, B-flow imaging, and CEUS in the assessment of intra-testicular tumor lesions in dogs. We assumed that such a multiparametric ultrasound approach would further improve the diagnostic and descriptive potential of testicular lesions, while also trying to evaluate malignancy as an increase in the vascularity of testicular lesions.

## 2. Materials and Methods

### 2.1. Case Selection

All dogs included in this prospective study were presented to the Tyrus Veterinary Clinic from January 2016 to May 2018. In all cases, we obtained informed owner consent prior to enrolling the dogs in the study. We reviewed results for 27 consecutive dogs with complete medical records and ultrasonographic evidence of testicular tumor lesions supported by histopathological diagnoses following orchiectomy. Clinical data included signaling, medical history, physical findings, as well as routine hematological and biochemical analyses. The diagnosis of a suspected testicular tumor-like lesion was based on initial clinical findings and/or ultrasonographic images. Once suspected, abdominal ultrasound examination and chest X-ray were also performed. After conventional B-mode ultrasound examination, all testicles were evaluated by color and spectral Doppler vascular ultrasonography, power Doppler, B-flow, and CEUS. After surgical intervention, all testicular lesions were examined histopathologically for a definitive diagnosis.

Exclusion criteria were the presence of concomitant pathologies, anamnesis of chronic diseases and/or acute illnesses in the last year, uncompleted diagnostic panel, or histopathological diagnosis not confirming the presence of a testicular tumor lesion.

### 2.2. Ultrasound Procedures and Image Analysis

All sonographic examinations were performed by the same operators using a LOGIQ E9 scanner (GE Medical Systems, Milwaukee, WI, USA) equipped with a (9L GE Medical Systems, Milwaukee, WI, USA) 2.5–8 MHz linear probe on unsedated dogs held in lateral recumbency. In dogs with multiple testicular lesions, we recorded data for each lesion. For all ultrasound methods used, the characteristics of the lesions were compared to normal surrounding testicular parenchyma, whilst in the case of lesions involving the whole gonad the normal parenchyma of the contralateral testicle was considered to compare data.

Longitudinal and transverse views were obtained from each testis using B-mode. B-Mode ultrasound parameters included anatomical site of affected testicle (scrotal or abdominal), maximal diameter (cm), location (left, right, or both testicle), margins (regular, irregular), echogenicity (hyperechoic, hypoechoic, isoechoic, anechoic or mixed), and echotexture (homogeneous or inhomogeneous) of the lesions relative to normal surrounding (or contralateral, in the case of a diffuse lesion) testicular parenchyma (Figure 1).

### 2.3. Color, Pulsed, and Power Doppler

The testicular artery was first detected in the pampiniform plexus by scanning the cranial pole of the testis. Then, the linear probe was moved towards the ventral region of testis to locate the marginal artery. Finally, the ultrasound beam was oriented to optimize the visualization of any lesions present and characterize their blood flow. Based on color Doppler imaging, the blood flow of the lesion was classified as either absent or present and, in such a case, as peripheral or intralesional (Figure 2A). Using pulsed wave mode, a sample volume was successively placed on the pampiniform plexus as well as marginal and lesion arteries (if present) to record the waveforms of at least three consecutive cardiac cycles. The PW was set as follows: frequency 3.1 MHz, pulse repetition frequency 3.3 kHz, and Gain 41 dB. The sample volume, the region where the vessel was studied, was positioned in the artery that allowed the best insonation angle when both peri and intralesional vessels were detected simultaneously in the same lesion. The insonation angle between the Doppler stream and the course of the vascular segment was manually aligned and the measured blood flow velocity was automatically corrected. The measurements with an angle >20° were disregarded. However, the measurements of three different waveforms in each vessel were taken and we made an average to reduce any errors. For each testicular artery, quantitative blood flow analysis included the evaluation of peak systolic velocity (PSV), end diastolic velocity (EDV), resistance index (RI), and pulsatility index (PI) (Figure 2B).

Using power Doppler, the blood flow of the testicular lesions was color mapped and classified as for color Doppler when present. Doppler signals were then quantified by a computer-assisted image analysis system using open-source software (ImageJ http://rsbweb.nih.gov/ij/, accessed on 15 November 2021) to measure the total number of color pixels.

### 2.4. B-Flow Examination

B-flow examination was focused of the vascularization on the testicular tumor lesions. Once testicular parenchyma was visualized by B-mode, the system was switched to B-flow. Still images and video clips were saved for each location examined for subsequent analyses. The vessels of the tumor lesions were scored as 0, 1, or 2 if absent, smaller, or larger than 2 mm in diameter, respectively. Moreover, the vascularization was classified as P if the vessels were perilesional or I if distributed within the tumor lesion (Figure 3).

### 2.5. Contrast-Enhanced Ultrasonography

We used the same linear transducer 9 L (GE Medical Systems, Milwaukee, WI, USA) set at a frequency of 2.5 MHz with a 0.09 mechanical index and a 90 dB dynamic range for CEUS evaluation. For each dog, a bolus dose (0.03 mL/kg of body weight) of the freshly prepared contrast agent (SonoVue^R^, Bracco Imaging, Milan, Italy) was rapidly infused via a three-way valve connected to an 18/20 G catheter in the cephalic vein. The contrast injection was immediately followed by a 5 mL saline flush (NaCl 0.9%), with repeated administrations of the contrast medium being used in cases of multiple lesions. All CEUS examinations were performed by the same two operators. The first operator injected the contrast medium through the catheterized vein, while the second performed the US scans by holding the transducer as still as possible at the selected position during the contrast study. This procedure was performed in both gonads with an interval of 5 min. During each examination, a 1.5 min video clip was recorded simultaneously with injection of the contrast agent and saved in the hard disk of the ultrasound machine for subsequent analyses.

Contrast enhancement patterns during the distribution phases were first evaluated qualitatively for each testicle to assess the following parameters compared to normal testicular parenchyma: (i) arrival time of contrast medium into the lesion (early or late); (ii) wash-in enhancement (homogeneous or inhomogeneous; (iii) intensity of contrast (hyperenhanced, isoenhanced, or hypoenhanced) (Figure 4).

Using the software integrated with the ultrasound machine, quantitative assessment of contrast enhancement was performed on specific regions of interest (ROIs). In particular, two ROIs of the same size (or more in case of multiple lesions) were drawn for each diseased testicle: one included the lesion (or the lesions) and the other the normal parenchyma. In the case of large lesions involving the whole gonad, the second ROI was drawn on normal parenchyma of the contralateral testicle. For each ROI, the software generated time–intensity curves to calculate the following parameters: peak intensity (PI, dB), i.e., the maximum intensity of the signal produced by the contrast; time-to-peak (TTP, s.), i.e., the interval from injection (time 0) until PI, and the area under the curve (AUC). Moreover, the washout (WO, s.) was arbitrarily defined as the time interval from TTP until signal intensity declined by 40% of PI (Figure 5).

### 2.6. Histopathological Examination

Testicles were submitted for histopathologic examination in 10% buffered formalin and were then trimmed, embedded in paraffin wax, and sectioned at 4 μm. The sections were stained with hematoxylin and eosin and examined for evidence of abnormal testicular parenchyma. Definitive diagnosis of the testicular lesions was made by histopathologic evaluation reviewed by a single veterinary pathologist blinded to the imaging ultrasound findings and clinical diagnosis.

### 2.7. Statistical Analysis

Two off-site radiologists (co-authors P.B. and A.T.) retrospectively reviewed the conventional ultrasonography techniques, Doppler, B-flow, and CEUS images of each dog used for the statistical analysis. Neither radiologist had performed the US examinations and both were blinded to the final diagnoses and to any other clinical information. Consensus was achieved for all variables examined and the final interpretation used for the statistical evaluation was based on the previously defined diagnostic criteria.

Discrete variables are presented as median and range, while categorical data as frequency or percent frequency. All quantitative data, evaluated for normality by the Kolmogorov–Smirnov test, are reported as mean ± standard deviation. The independent *t*-test was applied for comparing Doppler and CEUS quantitative parameters between testicular lesions of each different tumor type and normal parenchyma. The comparisons between ultrasonographic blood flow parameters of different testicular tumors were tested using ANOVA followed by post hoc Tukey test for contrast between means. For comparisons of categorical data, the chi-sqare test or Fisher exact test were applied. A *p* value <0.05 was considered indicative of a statistically significant difference.

Agreement between color Doppler, Power Doppler, and B-flow techniques for the evaluation of testicular tumor blood flow patterns was calculated using Fleiss kappa; 95% confidence intervals (CIs) are also provided. Kappa values were interpreted as follows: between 0.21 and 0.40 were considered as slight agreement, between 0.41 and 0.60 as moderate agreement, while values between 0.61 and 0.80 as substantial agreement. Their diagnostic sensitivity (true positives/(true positive + false negative)), specificity (true negatives/(true negative + false positive)), and positive and negative likelihood ratios (calculated as sensitivity/(1-specificity) and (1-sensitivity)/specificity, respectively) in differentiating the tumor types as well as their accuracy were assessed by comparing the findings with histological results. Based on our own clinical evidence, in the case of leydigomas, true positive cases were lesions characterized by the presence of peri/intralesional arteries and histological confirmation. True negative cases were lesions characterized by the presence of intralesional arteries and histological confirmation of tumors other than leydigoma. False positive lesions were those with the presence of peri/intralesional arteries and tumors other than leydigoma. False negative cases were defined as lesions with intralesional arteries and histological confirmation of a leydigoma. In the case of seminomas, the distinctive trait for a positive test was the presence of an intralesional artery and histological confirmation.

Demographic and specific characteristics of testicular tumors were reported descriptively for all the lesions in the study population. For comparative statistical analysis, whenever bilateral lesions of the same etiology were found, we included only the data of the right testicle. Whenever multiple lesions were present in the same testicle, we only used only values obtained from the greatest lesion among those of the same etiology. All statistical analyses were performed using commercially available software.

## 3. Results

### 3.1. Study Dogs and Reference Diagnosis

Twenty-seven dogs met the inclusion criteria. The dogs ranged in age from 5.5 to 16 years old (median, 11.0 years) and in weight from 5 to 35 kg (median, 26 kg). There were sixteen mixed breed (MB) dogs, two Doberman pinschers (DP), and one for each of the following breeds: American Stafforshire (AS), Basset Hound (BH), Dachshund (DH), Epagneul Breton (EB), French Bulldog (FB), German shepherd (GS), Italian Mastiff (IM), Labrador Retriever (LR) and Pointer (PR).

Twelve dogs had unilateral lesions (five on the right and seven on the left testis), while 15 dogs had bilateral lesions. In total, the 42 (20 right and 22 left) pathological testes had 46 tumor lesions, 38 of which were single, including one in a cryptorchid testis and four were multiple. Testicular lesions (Table 1) included 12 seminomas (26.1%), seven Sertoli cell tumors (15.2%), 21 Leydig cell tumors (45.7%), and six mixed tumors (13.0%). The mixed tumors were Leydig cell tumors combined either with Sertoli cell tumors (three lesions) or with seminoma (three lesions). By B-mode, we identified 45 of 46 lesions.

The seminomas (*n* = 12) were found in eight dogs. The tumors were unilateral in one dog involving the right testicle and bilateral in four dogs. In the other three dogs, the lesions (one left and two right) were associated with a different type of tumor in the contralateral testicle. The Leydig cell tumors (*n* = 21) were detected in 14 dogs. In six dogs, the lesions were unilateral involving the left (*n* = 5) and right (*n* = 1) testis. In four dogs, the leydigomas (one left and three right) were associated with a different type of tumor in the contralateral testicle. In the other four dogs, the leydigomas were bilateral with double lesions in three testicles. The sertoliomas (*n* = 7) were found in six dogs. One sertolioma was not detected using conventional B-mode, Doppler ultrasound, or B-flow imaging and was revealed only at CEUS. Four dogs had unilateral lesions, one on the left retained testicle and three on the right testis, one dog evidenced bilateral lesions, while in one dog the sertolioma were associated with a different type of tumor in the contralateral testicle. Mixed tumors (*n* = 6) were found in the testicles of five dogs. Only one lesion was unilateral on the left testicle, whilst the others appeared in bilateral lesions (with other non-mixed tumors found in contralateral testes).

No signs of metastasis were identified by ultrasound examination of abdominal lymph nodes or thoracic X-ray images, with the exception of two dogs. One dog showed the involvement of aortic lymph nodes (the cytological examination confirmed a metastasis of a bilateral sertoliomas) while the other had lung metastasis visible through X- ray due to bilateral seminomas.

### 3.2. Conventional Ultrasonography of Testicular Lesions

Testicular lesions included in the comparative analysis (*n* = 33) had a mean size of 16.0 ± 10.9 mm, 26 of which (78.8 %) were small enough to leave normal testicular parenchyma around the lesions (focal tumors), while seven (21.2%) occupied the whole testes (diffuse tumors). Eleven lesions (42.3%) were rounded and fifteen (57.7%) were oval-shaped. Out of 33 lesions, 10 (30.3%) were hyper-echogenic, one (3.0%) isoechoic, and the remaining 22 (66.7%) hypoechoic. In eight cases, cystic structures were evidenced within the lesions by conventional ultrasonography and confirmed histopathologically.

Seminomas (*n* = 8) appeared mainly hypoechoic (six in eight, 75.0%) and less frequently hyperechoic (two in eight, 25.0%), with either round (37.5%) or oval lesions (62.5%), mainly with irregular margins (seven in eight, 87.5%) and inhomogeneous pattern (five in eight, 62.5%). They ranged in size from 6.6 to 55 mm (19.1 ± 16.2 mm).

Leydigomas (*n* = 14) were all almost focal lesions (13 in 14, 92.9%), either round (eight in 14, 57.1%) or oval (six in 14, 42.9%), 4 to 30 mm in size (13.5 ± 8.4 mm). The lesions were mainly hypoechoic (10 in 14, 71.4%) and hyperechoic (four in 14, 28.6%) to the surrounding parenchyma with heterogeneous pattern in nine out of 14 (64.3%) and irregular margins in eight out of 14 cases (57.1%). In three lesions, anechoic cystic-like round structures were detected.

The sertoliomas (*n* = 6), ranging from 6 to 25 mm in diameter (16.2 ± 7.0 mm), were mainly focal lesions (five in six, 83.3%) showing an oval shape (four in six, 66.7%) with either well defined (three in six, 50.0%) or irregular (three in six, 50.0%) margins. They showed homogeneous (two in six, 33.3%) or inhomogeneous (four in six, 66.7%) patterns and hypo/isoechoic/hyperechoic structures in three, one, and two cases, respectively. In half of the lesions, conventional ultrasonography evidenced anechoic cystic-like round structures.

The mixed tumors (*n* = 5) were either focal (60.0%) or diffuse (40.0%) lesions, 5–30 mm in diameter (17.6 ± 11.4 mm). They were mainly oval shaped (four in five, 80.0%) with irregular margins (five in five, 100%) and an inhomogeneous pattern (four in five, 80.0%) with either hypo (three in five, 60.0%) or hyperechoic structures (two in five, 40.0%). In two lesions, there were anechoic cystic structures. In these mixed lesions, Leydig cell tumors were associated with seminoma (three cases) or with sertolioma (two cases).

The mean sizes of testicular lesions did not vary between the different tumor types (F = 0.4743, *p* = 0.703). Similarly, the type, shape, and margin of lesions as well as their echogenicity did not differ between tumor types.

### 3.3. Doppler Ultrasonographic Study

With color Doppler ultrasonography, arteries of the pampiniform plexus and marginal arteries of testis were easily visualized in all dogs. Using color Doppler ultrasound, the blood flow pattern was perilesional in eight tumors, intralesional in 15, and both perilesional and intralesional in five lesions, while there was no blood flow pattern observable in the remaining five lesions, including three leydigomas, one seminoma, and one sertolioma (Table 2). Counted jointly, perilesional and perilesional/intralesional patterns, the frequencies of blood flow patterns among testicular tumors varied significantly (chi-sqare (3, 28) = 9.938, *p* = 0.0191). The proportion of leydigomas with either a perilesional and/or perilesional/intralesional blood flow pattern was higher that of all the other tumors counted together (two in 11 vs. 13 in 17, *p* = 0.0056). This resulted in a sensitivity of 81.8%, a specificity of 76.5%, a correct classification rate of 78.6%, and a positive likelihood ratio of 3.5.

Among the pulsed-wave Doppler velocimetry values of pampiniform plexus (*n* = 33), PI and RI differed (*p* < 0.01) between the testicular tumors (Table 3). The PI of sertoliomas was higher than that of seminoma and mixed tumors (1.59 vs. 0.76 and 0.81, *p* = 0.0068 and *p* = 0.0123, respectively). The RI of seminomas was lower than that of sertoliomas (0.43 vs. 0.66, *p* = 0.0124). In the marginal artery of affected testicles (*n* = 33), PI and RI varied (*p* < 0.01) between the different tumor types. The mean values of both indexes were higher in sertoliomas (1.67 ± 0.59 and 0.67 ± 0.11, respectively) than in leydigomas (0.57 ± 0.22 and 0.40 ± 0.14, *p* = 0.00004 and *p* = 0.0016, respectively), seminomas (0.50 ± 0.23 and 0.40 ± 0.13, *p* = 0.00001 and *p* = 0.00009, respectively), and mixed tumors (0.84 ± 0.63 and 0.49 ± 0.19, *p* = 0.0006 and *p* = 0.0245). With this technique, lesion blood flow was sampled only in about half of the testicular tumors considered for the comparative analysis (16 in 33, 48.5%). In the intra- and/or perilesional arteries, PSV and EDV (Table 3) were higher in leydigomas (*n* = 7; 29.5 ± 10.0 and 19.1 ± 6.9, respectively) than in all the other tumors considered together (*n* = 9; 16.6 ± 3.6 and 8.6 ± 23, respectively).

In five testicular tumor lesions, power Doppler failed to detect any kind of vascularization, while it was perilesional in six tumors, intralesional in 11, and both perilesional and intralesional in the remaining 11 lesions (Table 4). Counted jointly, perilesional and perilesional/intralesional patterns, the frequencies of blood flow patterns varied significantly among testicular tumors (chi-sqare (3, 28) = 8.714, *p* = 0.0333). In leydigomas, the frequency of perilesional and/or perilesional/intralesional blood flow pattern (10 in 12) was higher (chi-sqare (1, 28) = 4.504, *p* = 0.0338) than that of all the other tumor types combined (7 in 16). This resulted in a sensitivity of 83.3%, a specificity of 56.3%, a correct classification rate of 67.9%, and a positive likelihood ratio of 1.9. Power Doppler signals of intralesional and perilesional vessels (Table 4) did not differ between the different testicular tumors, although their mean values were higher in leydigomas (*n* = 12; 10,392 ± 9655) than in seminoma (*n* = 6; 6192 ± 5486), sertoliomas (*n* = 5; 5519 ± 3127), and mixed cell tumors (*n* = 5; 5519 ± 5115).

### 3.4. B-Flow Examination

Using the B-flow technique, blood flows were easily imaged in testicles of all dogs and in all lesions, except one sertolioma. In the comparative analysis, the vascularization type of lesions was scored as one in 19 (57.6%) and as two in 13 (39.4%) out of 33 tumors. In one lesion (3.0%), vascularization was scored as 0 (Table 5). The frequencies of vascularization types did not vary among the different tumor types (chi-sqare (3, 33) = 3.438, *p* = 0.3294).

The vascular patterns of testicular lesions varied between the different types of tumors (chi-sqare (3, 33) = 12.491, *p* = 0.0058). In leydigomas, the frequency of perilesional and/or perilesional/intralesional blood flow pattern (12 in 14) was higher (chi-sqare (1, 33) = 11.507, *p* = 0.00317) than that of all the other tumor types (five in 19). In seminomas, the frequency of an intralesional blood flow pattern (seven in eight) was higher (chi-sqare (1, 33) = 6.435, *p* = 0.01118) than that of all the other tumor types considered together (Table 5). This resulted in a sensitivity of 85.7%, a specificity of 73.7%, a correct classification rate of 78.8%, and a positive likelihood ratio of 3.3.

### 3.5. Contrast-Enhanced Ultrasonography

No adverse side effects associated with the intravenous injection of the contrast agent were observed.

CEUS identified all testicular tumor lesions, including those not revealed by conventional ultrasonography, Doppler ultrasound, or B flow. Qualitatively, in the comparative analysis, 21 (63.6%) of the 33 lesions showed an inhomogeneous pattern during the wash-in, while 12 (36.4%) were homogeneous. The blood flow patterns of wash-in did not differ between tumor types (chi-sqare (3, 33) = 5.942, *p* = 0.1153). Eight of 33 lesions (24.2%) showed a hypoenhanced wash-in phase, whilst 18 (54.5%) were hyperenhanced and the remaining seven (21.2%) isoenhanced. The frequencies of wash-in contrast enhancement did not vary between leydigomas and the other tumor types (chi-sqare (2, 33 = 3.6192, *p* = 0.163723).

Quantitative perfusion parameters of time–intensity curves derived from ROIs positioned on testicular tumor lesions and normal testicular parenchyma were taken. Peak intensity and AUC mean values of leydigomas (*n* = 14) were higher (*p* < 0.05) compared to those of paired normal parenchyma (−52.1 ± 6.6 vs. −56.6 ± 4.3 and 542.3 ± 317.2 vs. 362.0 ± 210.0, respectively), while TTP was lower (29.7 ± 16.6 vs. 31.9 ± 18.2 s.). The peak intensity and AUC mean values of seminomas (*n* = 8) were higher (*p* < 0.01) compared to those of paired normal parenchyma (−52.1 ± 4.7 vs. −57.9 ± 3.3 and 406.4 ± 192.4 vs. 278.1 ± 168.0, respectively), while TTP was lower (*p* < 0.01; 26.9 ± 4.3 vs. 33.5 ± 8.4 s.). Peak intensity, TTP, and AUC mean values of sertoliomas and mixed cell tumors did not vary from those of corresponding normal parenchyma.

CEUS parameters (PI, TTP, AUC, and washout) derived from lesion ROIs did not vary among the different tumor types.

### 3.6. Agreement between US Techniques

Taking the three different US techniques (color Doppler, power Doppler, and B-flow) to be the raters, interrater ultrasonographic agreement, as assessed by Fleiss kappa, showed substantial agreement (0.64) in identifying intra/perilesional blood flow patterns (95% CI: 0.42, 0.86).

## 4. Discussion

To our knowledge, this is the first study to evaluate a multiparametric ultrasonographic approach for the diagnosis of testicular tumors in dogs. In principle, an accurate evaluation of the blood flow of testicular tumors should allow a better characterization of their etiological nature than that obtainable by conventional ultrasonography alone. Indeed, color Doppler, power Doppler, or B flow can be used to examine the vascular patterns of testicular tumor lesions sharing a good rate of agreement. B flow appears to be superior in terms of sensibility. On the other hand, CEUS allows the detection of all testicular tumors, including those not identified by the other ultrasonographic techniques used. Despite that, CEUS did not allow for the differentiation of any testicular tumor type.

In the present study, there was a higher prevalence of dogs with Leydig cell tumors in accordance with previous reports [24,25]. Conversely, in other recent reports [4,26,27], seminomas had the highest incidence. In our opinion, these differences are likely due to different criteria for enrolling dogs and the variable number of cases included in each study.

Conventional B-mode ultrasonography is considered a reliable, non-invasive technique, valuable for the detection of testicular tumors [8], even though their imaging features are not useful for the characterization of any particular tumor type [9]. In our study, 2D ultrasonography was able to spot testicular tumors in all dogs, except a Sertolioma in the testicle of a Basset hound. In agreement with previous studies [9,28], the ultrasound characteristics of tumors evaluated in grayscale, such as size, shape, and margins, were extremely variable, making their appearances inconsistent for differentiating any tumor type. In particular, there was a prevalence of focal lesions compared to the diffuse lesions occupying the entire testicle. The lesions showed a mainly hypoechoic structure with no peculiar characteristic that could differentiate them based on tumor type.

Color Doppler ultrasonography, pulsed wave spectral Doppler, and power Doppler applications are all reliable techniques for evaluating blood flow perfusion in canine reproductive tracts in different physiological and pathological conditions [11,29,30,31]. Color and pulsed Doppler ultrasound can assess not only the presence and direction of the blood flow, but also its dynamic parameters. Color and pulsed Doppler ultrasonography with a high-frequency transducer represent the imaging of choice for the initial identification of suspected neoplastic testicular lesions in human males [32]. When using pulsed Doppler ultrasonography, however, careful interpretation is required to evaluate blood flow velocity because such a parameter depends on the beam angle insonating the vessel. Conversely, this limiting factor does not influence resistive and pulsatility indices: the first reflects the resistance to blood flow caused by microvascular beds distal to the site of measurement, while the other quantifies the pulsatility or oscillations of the waveform during a cardiac cycle. For these reasons, PI and RI are widely used to assess testicular blood flow perfusion in humans and animals in physiological [12,13,33,34,35,36] and pathological conditions [11,37,38]. Compared to color and pulsed Doppler, power-Doppler imaging has a major sensitivity even for slow blood flow. In addition, power Doppler mode can improve the estimation of blood flow even in vessels with small diameters and weak blood flow because it is not influenced by the insonating angle [39]. However, despite the proven usefulness of Doppler ultrasonography, to our knowledge, there are only two papers regarding the use of these imaging techniques for the diagnosis of testicular tumors in dogs [11,31].

Color Doppler ultrasonography failed to identify perilesional and/or intralesional vascularization in five out of 33 testicular tumors included in the comparative analysis, but one should keep in mind that this technique has limited resolution and may fail to identify blood flow in small vessels and/or lesions with poor blood flow. In the present study, leydigomas had a perilesional or combined peri/intralesional distribution patterns in 81.2% of the cases, while the other types of tumors had a mainly intralesional vascularization pattern. However, the distinction between intra and/or perilesional vessels cannot differentiate among the other types of testicular tumors, such as seminomas, sertoliomas, or mixed cell tumors. Thus, the recognition of distinctive vascular features with color Doppler ultrasonography has quite limited usefulness for the differential diagnosis of testicular tumors in dogs being greatly influenced by their prevalence. Bigliardi et al. [31] reported similar findings using color Doppler imaging in dogs with different neoplastic lesions.

In our study, pulsed Doppler allowed the examination of pampiniform plexus in the testicles of all dogs. While peak systolic and end diastolic velocities of the pampiniform plexus did not vary between testicles with different tumors, the resistance index as well as the pulsatility index were significantly higher in testicles with sertoliomas than in testicles with seminoma. By pulsed Doppler, the marginal artery was sampled in all lesions included in the comparative analysis. Once again, peak systolic and end diastolic velocities of the marginal arteries did not vary between testicles with different tumors, but both resistance and pulsatility indexes in sertoliomas were higher than in the other types of tumors. Taken together, these findings are difficult to interpret given that both resistance and pulsatility indexes are ratios of two other primary variables (i.e., peak systolic velocity and end diastolic velocity) which did not vary between the different tumor types. Pulsed Doppler identified some perilesional and/or intralesional vascularization only in 16 out of 33 testicular tumors. Nevertheless, leydigomas were characterized by higher mean values of PSV and EDV than those of the other tumors considered together. These results suggest that Leydig cell tumors may influence the vascular architecture of the lesion and/or its functional characters. However, these findings should be taken with caution due to the small number of tumors with detectable vascularization, making comparisons between each tumor type poorly reliable. Based on the present data, therefore, the potential ability of pulsed Doppler in differentiating the nature of testicular tumors remains questionable mainly due to the poor resolution of this technique. Similar results were reported by Gunzel-Apel et al. [11] and Bigliardi et al. [31] in dogs. Gunzel-Apel et al. showed that all tumors examined (*n* = 5) had a different blood flow pattern compared with normal testes, but they did not demonstrate any difference between testicular tumors. Somewhat similar results were reported by Bigliardi et al. [31], since they found an increasing perfusion of neoplastic testicular parenchyma (that accounted for 40 in 50) of all testicular lesions examined, with significantly higher values in tumor lesions. Moreover, they described a 40–50% increase of the resistive index of neoplastic lesions compared to RI values of non-neoplastic lesions.

In our study, power Doppler allowed the visualization of blood flow in 28 out of 33 tumor lesions without any improvement in the detection limit compared to color Doppler. Regarding the vascular patterns of testicular tumors, power Doppler confirmed the observation recorded with color Doppler with a predominant perilesional or combined peri/intralesional distribution pattern in leydigomas and an intralesional vascularization pattern in the other types of tumors. Similarly, based on the characterization between intra and/or perilesional vessels, power Doppler cannot differentiate among the other types of testicular tumors such as seminomas, sertoliomas, or mixed cell tumors. Quantitatively, the number of color pixels detected by power Doppler did not vary between testicular tumors.

B-Flow is based on the GE-patented, digitally encoded ultrasound technique that was introduced in high-frequency transducers several years ago to digitally suppress unwanted signals (e.g., noise and tissue) and boost weak signals (e.g., blood echoes) to overcome some limitations of Doppler [40]. This technique visualizes blood flow echoes in gray scale imaging with different gray intensities according to the reflector’s speed and dynamics. This technology allows the direct visualization of intravascular flow echoes during real-time gray-scale ultrasonography. In addition, the technique is relatively simple to operate, with fewer parameters to manipulate than color Doppler. B-Flow imaging is very sensitive and in one study of placenta blood flow in women it allowed the visualization of a larger number of small (less than 5 mm) vessels compared to color Doppler [41]. In our study, B flow allowed the detection of the vascularization patterns in all the lesions considered for the comparative analysis. The vascular distribution pattern considering the intralesional vascularization alone against the perilesional alone or peri-intralesional combined varied significantly among the different tumor types. As for color and power Doppler, with B flow there was a predominant perilesional or combined peri/intralesional distribution pattern in leydigomas and an intralesional vascularization pattern in the other types of tumors. Similarly, based on the different patterns of intra and/or perilesional vascularization, B flow cannot differentiate among the other types of testicular tumors such as seminomas, sertoliomas, or mixed cell tumors. On the contrary, the distribution of frequencies of arterial sizes did not vary among the different tumor types. As far we know, this is the first study to examine the vascularization of testicular tumors in dogs with B flow. Quite surprisingly, in human medicine, the B Flow sonographic technique is not used very often in testicular tumor diagnosis, despite being much more sensitive than color Doppler.

No dog experienced adverse reactions to the injection of contrast agent. Once again, our clinical findings add further support to the safety of CEUS as documented by many previous studies [42,43,44,45]. As said, CEUS identified all tumors, including the sertolioma that was not detected by the other ultrasonographic techniques including B flow imaging. Independent of the testicular tumor type, in fact, at least one or more of the parameters characterizing the distribution and/or the pattern of the contrast within the lesion greatly differed from that of normal surrounding testicular parenchyma. In leydigomas and seminomas, mean quantitative parameters (PI, TtP, and AUC) derived from time–intensity curves of the lesion ROIs significantly differed from those obtained from normal parenchyma. Conversely, sertoliomas and mixed cell tumors did not show any such differences. However, the same quantitative parameters as well as the qualitative ones did not vary between the different tumor types. Therefore, based on our results, CEUS cannot be considered the diagnostic test of choice to differentiate the etiology of testicular tumors in dogs. These results are in agreement with those reported in human medicine literature [46,47] but partially disagree with those found in dogs by Volta et al. [7]. The latter authors, in fact, reported that hypo or isoenhanced testicular lesions with intralesional vessels were associated to seminomas, thus allowing a partial differentiation between testicular tumors.

All ultrasonographic techniques employed here (color and power Doppler, B flow, and CEUS) examined in real time the vascular perfusion in the same testicular tumors, each one with its own potential diagnostic potential due to technical features and underlying software. Interestingly, while CEUS did not contribute to the differential diagnosis of testicular tumors, it was highly effective in their detection by revealing an extra lesion. However, we cannot exclude that CEUS could have an even greater sensitivity in spotting small tumors. In fact, the marginal contribution of CEUS to the identification of testicular tumors may have been influenced by the enrolment criterion based only on the presence of a lesion identified in B-mode. Pulsed Doppler evidenced some differences in blood flow velocimetry of the pampiniform plexus, testicular arteries, and intralesional arteries between the different tumor types. However, in the latter case, diagnostic performance was greatly hampered by its poor sensibility in imaging blood flows, which made this technique inapplicable in almost 50% of testicular tumor lesions. There was some relationship between the presence or absence of intralesional and/or perilesional vascularity as assessed by Doppler or B-flow signals and the different types of tumors. Certainly, compared to color Doppler and power Doppler, B-flow was more effective in detecting blood vessels in and/or around testicular lesions, even if these ultrasonographic techniques showed a good rate of inter-agreement.

## 5. Conclusions

In conclusion, the present work indicates that testicular tumors of dogs have subtly different vascular patterns, a condition that could help in identifying their etiology. To this end, a multiparametric approach with the simultaneous use of color Doppler, B-flow, and CEUS ultrasonography is highly recommended. Despite the fact that our results are somewhat preliminary due to the limited number of dogs enrolled, we believe that these techniques could be useful to improve the detection of testicular tumors and to investigate clinical problems of hypo/infertility of male dogs. This would be particularly useful in those cases where a unilateral castration could be suggested, thus preserving future fertility, or where the presence of a tumor with low metastatic incidence may suggest skipping surgery and avoiding any anesthesiological risk.

## Figures and Tables

**Figure 1 animals-12-00210-f001:**
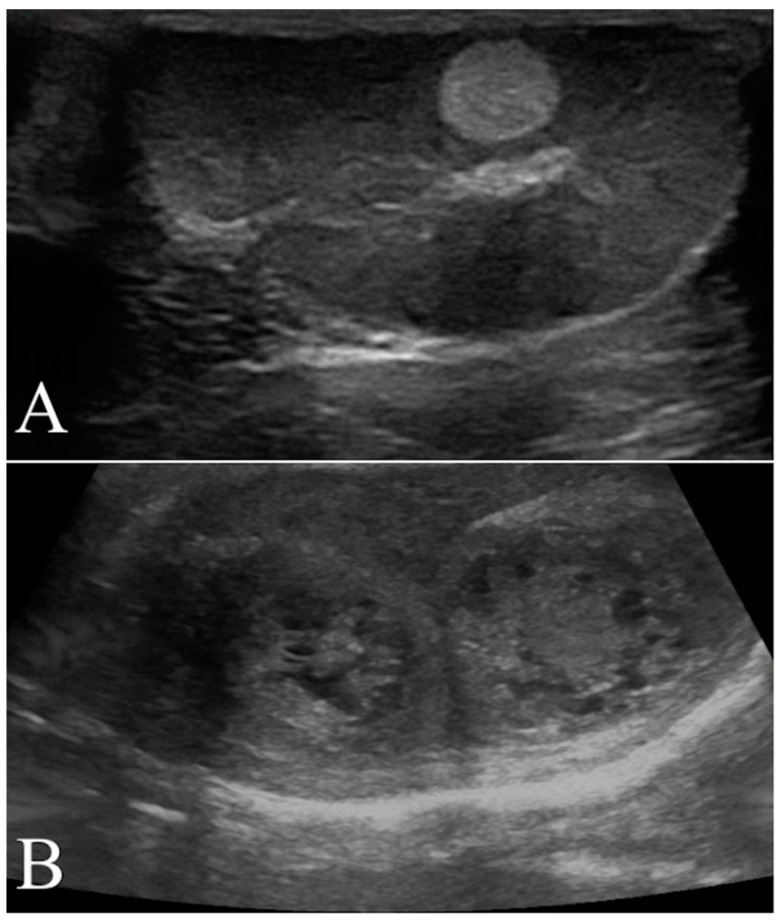
Representative B-Mode ultrasonography scans. (**A**) Left testis of a 11 year old mixed breed dog with a focal hyperechoic nodule (seminoma) characterized by homogenous echotexture and regular margin. (**B**) Mixed cell tumor (leydig and seminoma) recorded as two nodular lesions in the right testis of a 13 year old Epagneul Breton. The lesions had mixed patterns, inhomogeneous echotextures and irregular margins.

**Figure 2 animals-12-00210-f002:**
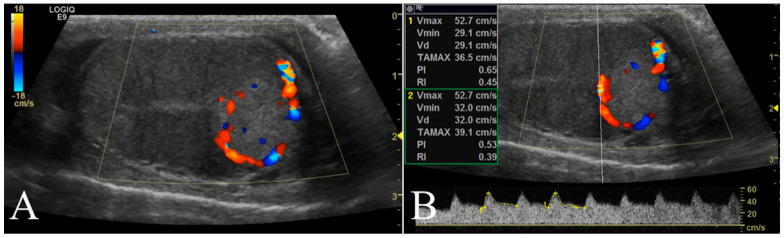
Representative Doppler characterization of a nodular lesion (leydigoma) found in the left testicle of a 11 year old Italian Mastiff. In particular, picture 2 (**A**), recorded by color Doppler, allowed the visualization of blood flow distributed peripherally to the lesion, while picture 2 (**B**) showed the same blood flow sampled by pulsed wave mode for quantitative analysis.

**Figure 3 animals-12-00210-f003:**
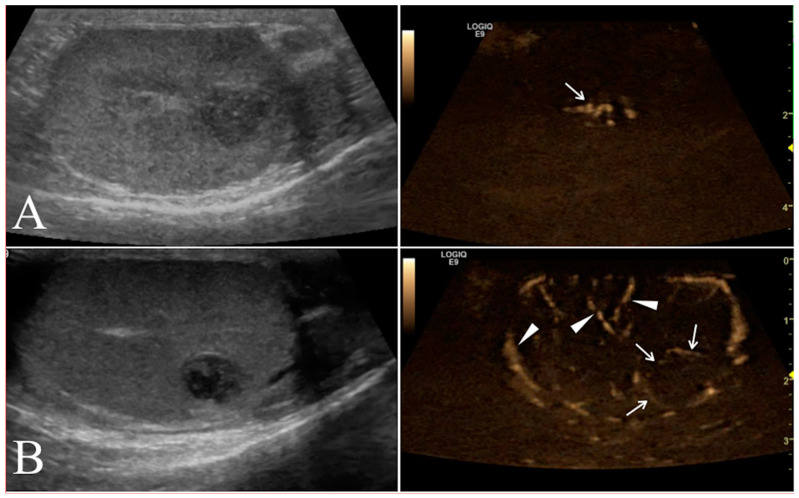
Conventional ultrasonographic scans (left panels) of two different seminomas found respectively in the right testicle of a 10 year old Labrador Retriever in (**A**) and in the right testicle of a 9.5 year old German shepherd in (**B**). In the right panel the B-flow images show the I distribution of the lesional vessels (arrow) scored as 2 in (**A**), while in (**B**) it was possible to record the P pattern of vessels distribution (arrows) scored in this case as 1. In this B-flow image, it is also possible to visualize some physiological testicular vessels (arrows head)**,** represented by the marginal region and intra-testicular branches of testicular artery.

**Figure 4 animals-12-00210-f004:**
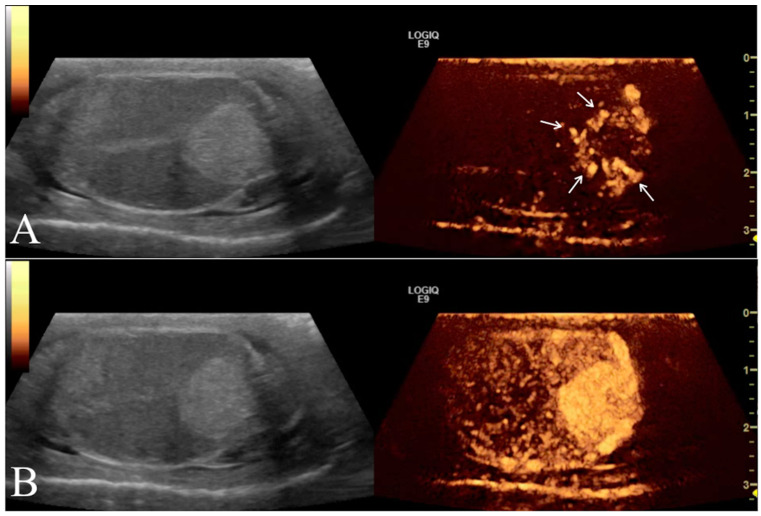
Left testicle leydigoma of a 11 year old Italian Mastiff (the same of Figure 2). The left panel shows the sagittal B-mode ultrasound highlighting a focal hyperechoic nodule with homogeneous echotexture and regular margin. Representative contrast-enhanced ultrasound images of different contrast distribution phases are represented in the right panels. After 16 s from contrast injection it is possible to appreciate an early distribution of the contrast within the lesions (Arrows) compared to the surrounding parenchyma (**A**). Later (25 s) the contrast medium distributes homogeneously within the lesion that remain hyperenhanced compared to the normal testicle (**B**).

**Figure 5 animals-12-00210-f005:**
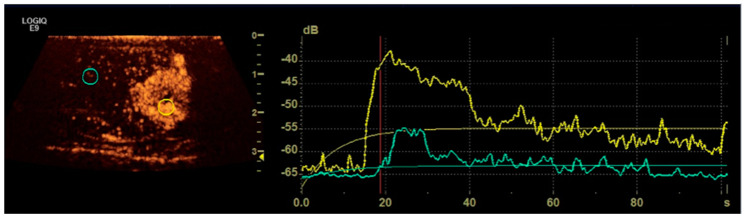
Representative image showing quantitative CEUS examination with ROIs positioned within the lesion (yellow ring) and in the normal parenchima (blue ring) respectively on the left and corresponding time-intensity curves in the right.

**Table 1 animals-12-00210-t001:** Characterization of the 46 testicular tumor lesions detected in 27 dogs of different breeds included in the study.

Testicular Distribution	Site	Tumor Type	Breeds *	Number of Lesions		
				Single	Multiple	Total
Monolateral	Right	Leydigoma	FB	1		5
		Sertolioma	2 MB + EB	3		
		Seminoma	GS	1		
	Left	Leydigoma	4 MB + DP	5		7
		Sertolioma	BH	1		
		Mixed	MB	1		
Bilateral	Right	Leydigoma	3 MB + IM + DP + AS + DH	7	2	17
		Sertolioma	PR	1		
		Seminoma	5 MB + LR	6		
		Mixed	MB	1		
	Left	Leydigoma	IM + DH + MB + DP + AS	5	1	17
		Sertolioma	MB + PR	2		
		Seminoma	4 MB + LR	5		
		Mixed	3 MB	3	1	
Total				42	4	46

* See text for codes of dog breeds.

**Table 2 animals-12-00210-t002:** Color Doppler characterization of arterial distribution in different tumor lesions (*n* = 33) included in the comparative analysis. The arteries (number and relative frequencies) were categorized as intralesional, perilesional, intra/perilesional, or absent.

Arteries	LEYDIG		SERTOL		SEMIN		MIXED	
	*n*.	%	*n*.	%	*n*.	%	*n*.	%
Intralesional	2	14	4	67	6	75	3	60
Perilesional	5	36	0	0	1	13	2	40
Intra/Perilesional	4	29	1	17	0	0	0	0
Absent	3	21	1	17	1	13	0	0
Total	14		6		8		5	

**Table 3 animals-12-00210-t003:** Pulsed-wave Doppler parameters derived from three different regions drawn within pampiniform plexus, marginal, and intra-testicular arteries of pathological testicles with different tumor lesions (*n* = 33) included in the comparative analysis. Peak systolic velocity = PSV, End diastolic velocity = EDV, Resistance index = RI, Pulsatility index = PI. Values are means ± S.D.

Arteries	Doppler Parameters	LEYDIGOMA			SERTOLIOMA			SEMINOMA			MIXED		
		Mean	±SD	*n*	Mean	±SD	*n*	Mean	±SD	*n*	Mean	±SD	*n*
Plexus	PSV	23.4	8	14	23	9.6	6	24.8	14.9	8	18.4	10.9	5
	EDV	9.9	4.2	14	6.4	2.8	6	11.8	3.8	8	9	7.5	5
	PI	1	0.3	14	1.6	0.8 ^a^	6	0.8	0.5 ^b^	8	0.8	0.2 ^b^	5
	RI	0.6	0.1	14	0.7	0.1 ^a^	6	0.4	0.2 ^b^	8	0.5	0.1	5
Testicular	PSV	23.4	13.7	14	23.3	0.6	5	18.1	5.2	8	18.5	10.5	4
	EDV	14.5	8.6	14	6.4	3.7	5	11.3	4.3	8	8.5	4	4
	PI	0.6	0.2 ^b^	14	1.7	0.6 ^a^	5	0.5	0.2 ^b^	8	0.8	0.6 ^b^	4
	RI	0.4	0.1 ^b^	14	0.7	0.1 ^a^	5	0.4	0.1 ^b^	8	0.5	0.2 ^b^	4
Intra/Peri lesional	PSV	29.5	10 ^a^	7	18.7	1.4 ^b^	4	13.1	3 ^b^	3	17.9	4.7 ^b^	2
	EDV	19.1	6.9 ^a^	7	10.3	2.4 ^b^	4	7.2	0.6 ^b^	3	7.5	2.3 ^b^	2
	PI	0.6	0.1	7	0.7	0.1	4	0.6	0.3	3	0.9	0.63	2
	RI	0.4	0.1	7	0.5	0.1	4	0.4	0.1	3	0.6	0.2	2

Different letters within the same row indicate significant differences for *p* ≤ 0.05.

**Table 4 animals-12-00210-t004:** Power Doppler characterization of arterial distribution in different tumor lesions (*n* = 33) included in the comparative analysis and color signals in pixels (mean values ± SD). The arteries (number and relative frequency) were categorized as intralesional, perilesional, intra/perilesional, or absent.

Tumor Type	Arterial Distribution								Power Doppler		
	Internal		Peripheral		Both		Absent		Pixel Values		
	*n*.	%	*n*.	%	*n*.	%	*n*.	%	Mean	±SD	*n*
LEYDIG	2	14	3	21	7	50	2	14	10,391	9655	14
SERTOL	4	67	0	0	1	17	1	17	5519	3127	6
SEMIN	4	50	1	13	1	13	2	25	6192	5265	8
MIXED	1	50	2	40	2	40	0	0	5519	5115	5

**Table 5 animals-12-00210-t005:** B-Flow characterization (numbers and relative frequencies) of arterial vascularization and arterial distribution in different tumor lesions (*n* = 33) included in the comparative analysis. The arterial vascularization was categorized as 0, 1 or 2 if absent, smaller or larger than 2 mm in diameter respectively; the arterial distribution was classified as P if the vessels were perilesional or I if distributed within the tumor lesion and I/P if both intra/perilesional.

Lesion Type	*n*.	Vascularization Codes						Distribution Flow Types					
		0	%	1	%	2	%	P	%	I	%	I/P	%
LEYDIG	14	0	0	7	50	7	100	6	43	2	14	6	43
SERTOL	6	0	0	4	67	2	33	0	0	4	67	2	33
SEMIN	8	1	13	6	75	0	16	1	13	7	88	0	0
MIXED	5	0	0	2	40	3	60	2	40	3	60	0	0

## Data Availability

The data presented in this study are available on giustified request from the corresponding author.

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
