# Peer review of "Characterization of Testicular Tumor Lesions in Dogs by Different Ultrasound Techniques"

_animals, 2022, doi:10.3390/ani12020210_

Round 1

Reviewer 1 Report

interesting paper, great content

Testicular tumors represent the most common tumors of the canine male genitalia with a prevalence that ranges from 2 to 60% and an incidence that increases with age Among the testicular tumors, Seminoma, Sertoli cell tumor, and Leydig cell tumor are the most common

Since many years, two-dimensional ultrasonography is part of the routine investigation of the male reproductive tract in both human and veterinary medicine. It represents the imaging technique of choice for the detection and characterization of testicular lesions. Despite this diagnostic approach can help in differentiating neoplastic processes from other pathologies such orchitis, testicular torsion, and epididymitis, the ultrasonographic changes are not specific enough to identify between different types of testicular tumors . In fact, the ultrasonographic features of testicular tumors are extremely variable.

Therefore, once detected, testicular tumor-like lesions require further characterization. Recently, color Doppler ultrasonography and contrast enhanced ultrasound (CEUS) have been applied in men and dogs to improve the diagnostic accuracy of testicular tumors and help to decide whether sparing surgery or total orchiectomy was indicated. In fact, whereas two-dimensional ultrasonography offers valuable, but only structural information about testicular tumors, color Doppler and CEUS provide information about their real time blood flow. This information is clinically relevant because vascularization and micro-vessel density, i.e. the number of vessels per mm, reflect tumor angiogenesis and are related to its malignancy. In fact, many studies have shown that angiogenesis promotes tumor growth by supplying essential oxygen and nutrients to neoplastic cells. At the end of the last decade, the development of B-flow technique with high-frequency transducers allowed a non-Doppler visualization of blood flow imaging that was considered superior for evaluating complex hemodynamics of peripheral vascular pathologies in men.

Not surprisingly, therefore, a multiparametric approach consisting of conventional ultrasonography, color Doppler ultrasound, and CEUS, is increasingly being used in humans for differentiating benign from malignant intra-testicular lesions. On the contrary, as far we know, such a multiparametric approach has never been tested in the small animal practice although, especially in stud dogs, the use of reliable diagnostic tools would be relevant for the detections and characterization of testicular tumors that can compromise fertility. In addition, an early diagnosis of a testicular tumor can also reduce the risk of metastasis that occurs in 10% to 20% of dogs.

Therefore, in the present study was evaluated the diagnostic accuracy of two-dimensional ultrasonography combined with Doppler ultrasound, B-flow imaging, and CEUS in the assessment of intra-testicular tumor lesions in dogs. We assumed that such a multiparametric ultrasonographic approach would further improve the diagnostic potential given that previous reports on both humans and dogs have linked malignancy with increased vascularity of testicular lesions.

Author Response

Dear Reviewer, Thank you for reading our manuscript and for appreciating it

Reviewer 2 Report

The article presents an adequate hypothesis and a description of materials and methods that is replicable. The results are concrete and provide precise recommendations to perform an adequate ultrasonographic in dogs with testicular pathologies. The images provided by the authors are adequate. The discussion is adequate.

In conclusion, the article provides important results in the field of ultrasonography by presenting suggestions and recommendations for an adequate technique to determine testicular pathologies in dogs.

The main question addressed by the research is to differentiate diverse testicular tumors in the dog by distinctive ultrasonographic techniques.

It is a relevant issue since it could contribute to determining the etiology based on the different vascular patterns.

The present work implements a multiple ultrasonographic evaluation consisting of conventional ultrasonography, color Doppler ultrasound and contrast enhanced ultrasound in dogs with testicular tumors.

The results are concrete and provide precise recommendations to perform an adequate ultrasonographic in dogs with testicular pathologies.

It would be interesting to increase the number of animals in further studies.

The images provided by the authors are adequate

The conclusions are consistent with the evidence presented. References are adequate in number and individual contributions.

In conclusion, the article provides important results in the field of ultrasonography by presenting suggestions and recommendations for an adequate technique to determine testicular pathologies in dogs.

Author Response

Dear Reviewer, Thank you for reading our manuscript and for appreciating it.

The manuscript was reviewed by a native English speaker.

Reviewer 3 Report

Dear authors, 

Thank you for writing this comprehensive manuscript and all the work behind such a study. Your results are promising. However, I agree that other studies are warranted to further expand our knowledge about testicular neoplasia.

I have found some inaccuracies and vulnerable parts provided below, together with my comments. 

Page 7, line 208: I think the unit "mm" should be "micrometres".

Page 7, line 213: Which software did you use to run the statistical analyses? Who has performed and interpreted the statistical analyses?

Page 7, line 220: Breed is not a discrete variable but a qualitative (syn. nominal, categorical).

Page 7, line 223: Which type of t-test did you use?

Page 7, line 226: I know ANOVA and the post-hoc Tukey test. This part is clear to me. However, I don't understand the following part of the sentence: "... for contrast between means when significant." Would you please rephrase or explain it? 

Page 7, line 227: I hope the letter "c" is a typo or a machine error due to some text conversions. If you cannot type the greek letter "chí", you can write it like "chi-square test". Please, change consistently in the entire manuscript.

Page 7, lines 229-230: This is not a sufficient description of the planned comparison. I suspect that the blood flow pattern is an ordinal variable. The most appropriate method for agreement analysis for ordinal variables when more than two observers are compared (here: the ultrasonographic techniques) is Fleiss' Kappa. In the case of a continuous variable, the intra-class coefficient or Bland-Altman plot is the most suitable method. 

Page 7, lines 233-234: Sensitivity and specificity are not correctly defined. The correct formulas are below. Because of this inaccuracy, I don't trust some of the results. 

Sensitivity: true positive/(true positive+false negative)

Specificity: true negative/(true negative+false positive)

Pages 7 and 8, lines 237-244: Is this arbitrarily defined or based on any evidence? Would you please provide any references?

Page 10, lines 321-324: The numbers in the text are different from the numbers in Table 2. 

Page 10, lines 324-326: This sentence is unclear. Please, rephrase. 

Page 10, line 350: "Pulsed" should be "Pulsed-wave".

Page 11, lines 359-360: Why exactly did you combine these two variables? Have you also tried the other combinations of variables? Seems like a data dredging. 

Page 13, lines 422-426: The results presented in this chapter are unclear. The interrater agreement should indicate an agreement between two persons. I think, in this case, you compare two methods. The agreement analysis in both the chapters, Materials and Methods and the Results, is unclear. Would you please rephrase them? 

Page 13, line 461: Please, change the order of the words "pulsatility and resistive" according to the following sentence.

Author Response

Dear Reviewer, thank you for reading our manuscript so carefully and for all your suggestions, corrections and ideas. We are sure that your comments led to a significant improvement of the manuscript. Thank you!!
In the following, we refer to your comments and answer them:

Page 7, line 208: I think the unit "mm" should be "micrometres".

            Corrected

Page 7, line 213: Which software did you use to run the statistical analyses? Who has performed and interpreted the statistical analyses?

Added. The roles played by each author are listed in the Author Contribution Section.

Page 7, line 220: Breed is not a discrete variable but a qualitative (syn. nominal, categorical).

            Corrected

Page 7, line 223: Which type of t-test did you use?

            We used the independent Samples t test

Page 7, line 226: I know ANOVA and the post-hoc Tukey test. This part is clear to me. However, I don't understand the following part of the sentence: "... for contrast between means when significant." Would you please rephrase or explain it? 

            We simply meant that the post-hoc test was performed only when the ANOVA analysis indicated a significant difference between the means.

Page 7, line 227: I hope the letter "c" is a typo or a machine error due to some text conversions. If you cannot type the greek letter "chí", you can write it like "chi-square test". Please, change consistently in the entire manuscript.

            Yes, an error due to PDF conversion. Corrected throughout the text

Page 7, lines 229-230: This is not a sufficient description of the planned comparison. I suspect that the blood flow pattern is an ordinal variable. The most appropriate method for agreement analysis for ordinal variables when more than two observers are compared (here: the ultrasonographic techniques) is Fleiss' Kappa. In the case of a continuous variable, the intra-class coefficient or Bland-Altman plot is the most suitable method. 

We changed the statistic using the Fleiss kappa as indicated to assess the agreement among the Color Doppler, Power Doppler and B-flow US

Page 7, lines 233-234: Sensitivity and specificity are not correctly defined. The correct formulas are below. Because of this inaccuracy, I don't trust some of the results. 

Sensitivity: true positive/(true positive+false negative)

Specificity: true negative/(true negative+false positive)

            We agree: we simply oversimplified the concept, but data were handled correctly.

Pages 7 and 8, lines 237-244: Is this arbitrarily defined or based on any evidence? Would you please provide any references?

The parameters here considered for calculating sensitivity and specificity of color Doppler and B flow techniques were arbitrarily defined based on our own clinical evidences.

Page 10, lines 321-324: The numbers in the text are different from the numbers in Table 2. 

            Corrected

Page 10, lines 324-326: This sentence is unclear. Please, rephrase. 

Rephrased as suggested: “Counted jointly, perilesional and perilesional/intralesional patterns, the frequencies …”

Page 10, line 350: "Pulsed" should be "Pulsed-wave".

            Corrected

Page 11, lines 359-360: Why exactly did you combine these two variables? Have you also tried the other combinations of variables? Seems like a data dredging. 

We combined these two variables to avoid cells with “zero” values. At the same time, combining lesions with perilesional arteries with those having perilesional/intralesional arteries makes sense in identifying those lesions that had only intralesional arteries.

Page 13, lines 422-426: The results presented in this chapter are unclear. The interrater agreement should indicate an agreement between two persons. I think, in this case, you compare two methods. The agreement analysis in both the chapters, Materials and Methods and the Results, is unclear. Would you please rephrase them? 

            We have rephrased the whole paragraph accordingly to changes made in Materials and Methods

Page 13, line 461: Please, change the order of the words "pulsatility and resistive" according to the following sentence.

            Changed

Reviewer 4 Report

27th December

Characterization of testicular tumor lesions in dogs by different ultrasound techniques

General comments

In the present work, authors investigated the hypothesis that testicular tumor lesions could present specific vascular pattern, which could be detected and differentiated by ultrasonography. Testicular tumors are common in old dogs, however, their malignity is low. Diagnosis is reached by clinical findings and confirmed by ultrasonography. The management of testicular tumors is surgery; bilateral orchiectomy, without delay. The specific diagnosis is done later by histopathologic analysis of the lesions. The fact that the tumor type could be detected previously could had some additional and important information to the case. With appropriate clinical screening of middle aged to old dog, testicular tumors are detected previously to installation of several clinical consequences. A recent work by Nascimento et al (2020) done in 190 dogs, 220 neoplasms: Pesq. Vet. Bras. 40 (7) • July 2020 • https://doi.org/10.1590/1678-5150-PVB-6615, which stablishes the frequency of histology subtypes, confirms “it was observed that malignancy characteristics were uncommon”. So, the context for realizing the present work, the aims and the comparison with Humans, have to be rebuilt, as malignancy in dogs testicular tumors is not so common and comparison with human medicine is not so linear and parallel. Lines L70 to L83 have to rebuild, focusing the aims in other hypothesis.

Specific topics

L 105. In case that one testicle has multiple lesions, how were US-findings connected to histopathological?

L 117. Conventional ultrasonographic could be replaced by B-Mode ultrasonography?

L 122. Were the intratesticular arteries pulse detected and measured as well?

L 132. When authors refer to “lesional vessels”, to which testicular vessels are they specifically talking about? Please add that information. If it concerns a branch of testicular artery, it should be stated.

L 133. Please refer the settings of the equipment used in the present work for pulsed-Doppler analysis, including insonation angle.

L 138. The cursor for quantitative blood analysis should be placed in the middle of the artery, not in the periphery, as presented. Please verify and refer to the technique used.

L 161. Please specify which “physiological testicular vessels” are authors talking about.

Remaining methods used in the present work to detect and characterize testicular lesions are repeatable and robust enough.

L 213. Statistical analysis is adequate and solid, however with such “n”, it was expected a non-normal distribution.

L 260. The frequency of the different testicular tumors is not the same then in Summary. Please verify.

L 427 Discussion

Even without secondary effects, the use of CEUS in the testicular lesions context and clinical management is questionable when analysing the testicular lesions.

Discussion section is somewhat lengthy, but considering the variety of methods analysed, seems adequate.

References

In some references, authors have the volume and the issue, in others only the volume. Please, uniformize the style.

Author Response

Dear Reviewer,
thank you very much for reading our manuscript and the suggestions and corrections. In the following, we refer to your comments and answer them:

General comments

In the present work, authors investigated the hypothesis that testicular tumor lesions could present specific vascular pattern, which could be detected and differentiated by ultrasonography. Testicular tumors are common in old dogs, however, their malignity is low. Diagnosis is reached by clinical findings and confirmed by ultrasonography. The management of testicular tumors is surgery; bilateral orchiectomy, without delay. The specific diagnosis is done later by histopathologic analysis of the lesions. The fact that the tumor type could be detected previously could had some additional and important information to the case. With appropriate clinical screening of middle aged to old dog, testicular tumors are detected previously to installation of several clinical consequences. A recent work by Nascimento et al (2020) done in 190 dogs, 220 neoplasms: Pesq. Vet. Bras. 40 (7) • July 2020 • https://doi.org/10.1590/1678-5150-PVB-6615, which stablishes the frequency of histology subtypes, confirms “it was observed that malignancy characteristics were uncommon”. So, the context for realizing the present work, the aims and the comparison with Humans, have to be rebuilt, as malignancy in dogs testicular tumors is not so common and comparison with human medicine is not so linear and parallel. Lines L70 to L83 have to rebuild, focusing the aims in other hypothesis.

Thank you for this comment. Based on it, we have partly refocused the aims

Specific topics

L 105. In case that one testicle has multiple lesions, how were US-findings connected to histopathological?

            The testicles sent for histological examination were accompanied by an ultrasound description of the lesions indicating their number, size and orientation

L 117. Conventional ultrasonographic could be replaced by B-Mode ultrasonography?

            Changed as requested

L 122. Were the intratesticular arteries pulse detected and measured as well?

            We evaluated the pampiniform plexus, the marginal artery and the intratesticular artery linket to the lesions. We don’t evaluate other intratesticular artery  because their changes in literature where reported to be more associated to inflammatory process than neoplasia

L 132. When authors refer to “lesional vessels”, to which testicular vessels are they specifically talking about? Please add that information. If it concerns a branch of testicular artery, it should be stated.

            We refer to the vessels in the immediate periphery of the lesion or within the tumor parenchyma itself. These vessels can be intraparenchymal branches of the testicular artery, but also can have a neoangiogenic derivation linked to neplasia

L 133. Please refer the settings of the equipment used in the present work for pulsed-Doppler analysis, including insonation angle.

            Changed as requested

L 138. The cursor for quantitative blood analysis should be placed in the middle of the artery, not in the periphery, as presented. Please verify and refer to the technique used.

            In our opinion, the sampling is representative of the analyzed artery, perhaps being a still image it may not be perfectly aligned. However, the measurements of three different waveforms in each vessel were taken and we made an average to reduce any errors

L 161. Please specify which “physiological testicular vessels” are authors talking about.

            Represented by  the marginal region and intra-testicular branches of testicular artery

Remaining methods used in the present work to detect and characterize testicular lesions are repeatable and robust enough.

L 213. Statistical analysis is adequate and solid, however with such “n”, it was expected a non-normal distribution.

L 260. The frequency of the different testicular tumors is not the same then in Summary. Please verify.

            Corrected. Thank you for spotting these mistakes

L 427 Discussion

Even without secondary effects, the use of CEUS in the testicular lesions context and clinical management is questionable when analysing the testicular lesions.

In the balance of pro and cons, please consider that CEUS is extremely valuable in detecting small testicular tumors

Discussion section is somewhat lengthy, but considering the variety of methods analysed, seems adequate.

References

In some references, authors have the volume and the issue, in others only the volume. Please, uniformize the style.

            Corrected as needed

Reviewer 5 Report

Reviewer comments for manuscript ID animals-1485126 entitled ‘Characterization of testicular tumor lesions in dogs by different ultrasound techniques’

General comments

A unique study on one of the most common tumors in male dogs with a thorough diagnostic approach utilizing different types of ultrasonographic modalities. Despite the relatively fewer number patients used in the study, the authors have thoroughly studied various types of testicular tumors. I am also impressed with the statistical analysis of the data. This further enhanced the robustness of the study. Results and discussions stand out. I am hopeful that this study will provide a baseline for further research on this topic of clinical importance.

I have very few and minor corrections/suggestions for the authors.

Simple summary and abstract have the same content. Please read the journal instructions about the same and rewrite.

In the summary and abstract, I presume that Color Doppler Ultrasonography was done on the testes in intact dogs and then retrospectively the scans were reassessed following the histopathology of the tumors. If I am right, please clarify it for more clarity to the readers.

Specific comments

Line 27: What do you mean by ‘consecutive’ here? Please clarify or delete.

Line 51: Please replace ‘Despite’ with ‘though’

Line 60: Please delete ‘whereas’

Line 70: Please delete ‘Not surprisingly’

Line 73: Please replace ‘as far we know’ with ‘as per our knowledge’

Lines 73-76: Too long sentence. Please break it into two sentences.

Line 100: Please replace ‘uncompleted’ with ‘incomplete’

Line 129: Please clarify ‘sample volume’ in this context for the clarity of the reader.

Lines 252-63: Breed wise histopathology results in the table can be a valuable information. If possible, please include in the table.

Line 331 & table 2: The percentages should be in whole numbers and not decimals. Please check.

Table 4: The percentages should be in whole numbers and not decimals. Please check. Please check in further tables also.

Line 334: Please replace ‘sensibility’ with ‘sensitivity’

Line 585-88: Please reframe ‘In conclusion, the present work indicates that testicular tumors of dogs have vascular patterns subtly different, a condition that could help in identifying their etiology; to this end, a multiparametric approach with the simultaneous use of color Doppler, B-flow, and CEUS ultrasonography is highly recommended’ as ‘The present work reveals that testicular tumors of dogs have subtly different vascular patterns. These architectural details enhanced by a multiparametric approach with the simultaneous use of color Doppler, B-flow, and CEUS ultrasonography is highly recommended for identification of etiology of these tumors’

Line 594: Please add after these lines’… avoid any anesthesiological risk’ especially in geriatric subjects.

Author Response

Dear Reviewer,

We have accepted all your suggestions and modified the manuscript accordingly. We want to thank the efforts spent on reviewing our work:

Simple summary and abstract have the same content. Please read the journal instructions about the same and rewrite.

We have partly rewrote the summary in compliance with the suggestion indicated

In the summary and abstract, I presume that Color Doppler Ultrasonography was done on the testes in intact dogs and then retrospectively the scans were reassessed following the histopathology of the tumors. If I am right, please clarify it for more clarity to the readers.

            Clarified as requested

Specific comments

Line 27: What do you mean by ‘consecutive’ here? Please clarify or delete.

By “consecutive”, we means that, in the given time lapse, no dog was purposely removed from the study unless for reasons specified in the exclusion criteria. This procedure warrants statistical independence of cases under study.

Line 51: Please replace ‘Despite’ with ‘though’

            Changed as requested

Line 60: Please delete ‘whereas’

            Deleted as indicated

Line 70: Please delete ‘Not surprisingly’

            Deleted

Line 73: Please replace ‘as far we know’ with ‘as per our knowledge’

            Changed as requested

Lines 73-76: Too long sentence. Please break it into two sentences.

            Added a double sentence

Line 100: Please replace ‘uncompleted’ with ‘incomplete’

            Replaced

Line 129: Please clarify ‘sample volume’ in this context for the clarity of the reader.

            Clarify as requested

Lines 252-63: Breed wise histopathology results in the table can be a valuable information. If possible, please include in the table.

            We have included this information in Table 1.

Line 331 & table 2: The percentages should be in whole numbers and not decimals. Please check.

            Corrected as requested

Table 4: The percentages should be in whole numbers and not decimals. Please check. Please check in further tables also.

            Corrected as indicated

Line 334: Please replace ‘sensibility’ with ‘sensitivity’

            Replaced

Line 585-88: Please reframe ‘In conclusion, the present work indicates that testicular tumors of dogs have vascular patterns subtly different, a condition that could help in identifying their etiology; to this end, a multiparametric approach with the simultaneous use of color Doppler, B-flow, and CEUS ultrasonography is highly recommended’ as ‘The present work reveals that testicular tumors of dogs have subtly different vascular patterns. These architectural details enhanced by a multiparametric approach with the simultaneous use of color Doppler, B-flow, and CEUS ultrasonography is highly recommended for identification of etiology of these tumors’

            Thanks for the suggestion that we appreciate.

Line 594: Please add after these lines’… avoid any anesthesiological risk’ especially in geriatric subjects.

            Added as indicated